# Detection of pneumonia in children through chest radiographs using artificial intelligence in a low-resource setting: A pilot study

**Taofeeq Oluwatosin Togunwa**[1,2]*, **Abdulhammed Opeyemi Babatunde**[1,2], **Oluwatosin Ebunoluwa Fatade**[3], **Richard Olatunji**[3], **Godwin Ogbole**[3], **Adegoke Falade**[4]

**1** College of Medicine, University of Ibadan, Ibadan, Nigeria, **2** College Research and Innovation Hub, Ibadan, Nigeria, **3** Department of Radiology, University College Hospital, Ibadan, Nigeria, **4** Department of Paediatrics, University College Hospital, Ibadan, Nigeria

* togunwataofeeq@gmail.com

## Abstract

Pneumonia is a leading cause of death among children under 5 years in low-and-middle-income-countries (LMICs), causing an estimated 700,000 deaths annually. This burden is compounded by limited diagnostic imaging expertise. Artificial intelligence (AI) has potential to improve pneumonia diagnosis from chest radiographs (CXRs) through enhanced accuracy and faster diagnostic time. However, most AI models lack validation on prospective clinical data from LMICs, limiting their real-world applicability. This study aims to develop and validate an AI model for childhood pneumonia detection using Nigerian CXR data. In a multi-center cross-sectional study in Ibadan, Nigeria, CXRs were prospectively collected from University College Hospital (a tertiary hospital) and Rainbow-Scans (a private diagnostic center) radiology departments via cluster sampling (November 2023–August 2024). An AI model was developed on open-source paediatric CXR dataset from the USA, to classify the local prospective CXRs as either normal or pneumonia. Two blinded radiologists provided consensus classification as the reference standard. The model's accuracy, precision, recall, F1-score, and area-under-the-curve (AUC) were evaluated. The AI model was developed on 5,232 open-source paediatric CXRs, divided into training (1,349 normal, 3,883 pneumonia) and internal test (234 normal, 390 pneumonia) sets, and externally tested on 190 radiologist-labeled Nigerian CXRs (93 normal, 97 pneumonia). The model achieved 86% accuracy, 0.83 precision, 0.98 recall, 0.79 F1-score, and 0.93 AUC on the internal test, and 58% accuracy, 0.62 precision, 0.48 recall, 0.68 F1-score, and 0.65 AUC on the external test. This study illustrates AI's potential for childhood pneumonia diagnosis but reveals challenges when applied across diverse healthcare environments, as revealed by discrepancies between internal and external evaluations. This performance gap likely stems from differences in imaging protocols/equipment between LMICs and high-income settings. Hence,

**Data availability statement:** The imaging data collected during the study has been de-identified and available open source at https://doi.org/10.5281/zenodo.14185822.

**Funding:** This work was supported by the Duke Center for Policy Impact in Global Health (CPIGH) through a Mentored Student Research Grant on Children's Health Disparities to T.O.T. and A.O.B. The source of the grant was a CPIGH discretionary fund given to CPIGH faculty, established by the Duke Global Health Institute's Director. The funder had no role in the design or conduct of the study; design, collection, analysis, or interpretation of the data; or decision to submit the manuscript for publication. Additional information about the CPIGH is available at https://centerforpolicyimpact.org/.

**Competing interests:** The authors have declared that no competing interests exist.

public health priority should be developing robust, locally relevant datasets in Africa to facilitate sustainable and independent AI development within African healthcare.

## Author summary

Pneumonia is a leading cause of death in children under five, especially in low-resource settings like Nigeria, where access to diagnostic tools and expertise is limited. Our study explores how artificial intelligence (AI) can help address this gap by detecting pneumonia from chest X-rays. We trained an AI model using a large dataset of children's X-rays from the United States and tested it on images collected in Nigeria.

While the AI model performed well on the U.S. data, its accuracy dropped significantly when tested on the Nigerian X-rays. This reveals how differences in imaging techniques and equipment between countries can affect the performance of such models. It highlights the need for AI systems to be adapted to local contexts to ensure they are reliable and effective in real-world settings.

Our findings underline the importance of creating high-quality, locally relevant datasets in Africa to support the development of AI tools that address the unique challenges of the region. By investing in such efforts, we can improve access to life-saving technologies, particularly for vulnerable populations in resource-limited healthcare systems.

## Introduction

Globally, pneumonia remains a leading cause of under-5 mortality [1]. In 2015, approximately 700,000 under-5 children died from pneumonia worldwide, mostly in low and middle income countries (LMICs) [2] Sub-Saharan Africa (SSA) continues to bear a significant portion of this burden, with Nigeria accounting for 169,000 under-5 community-acquired pneumonia deaths in 2021, the highest absolute number for any single country [3]. Nigeria is not on track to achieve the Global Action Plan for Pneumonia and Diarrhoea targets to reduce mortality to 3 per 1,000 by 2025 [4]. This highlights the need for urgent, effective, evidence-based interventions.

Childhood pneumonia in LMICs is largely driven by viral pathogens like respiratory syncytial virus (RSV). However, bacterial pathogens contribute remarkably to moderate to severe disease, particularly in children with underlying health conditions [5]. Diagnosis in sub-Saharan Africa (SSA) heavily relies on clinical judgment [6]. Additional danger signs like feeding difficulties, convulsions and hypothermia categorize severe cases. While most children recover fully, around 3–5% experience pulmonary or systemic complications, contributing to significant morbidity and mortality [7]. The economic impact in Nigeria, though not precisely quantified, likely mirrors findings

from Uganda where almost 40% of households face catastrophic health expenditures due to childhood pneumonia treatment costs [8].

Standard diagnostic protocols for childhood pneumonia in LMICs typically involve clinical assessment supplemented by laboratory and microbiological testing. However, imaging becomes essential if outpatient treatment fails, or in cases of hospital admission or hospital-acquired pneumonia [9]. Chest radiographs are the primary imaging method for childhood pneumonia; nonetheless, significant variability in interpretation among radiologists is a common limitation of this technique [9]. Moreover, paediatric radiology in LMICs faces additional challenges: scarcity of specialized equipment, radiation safety concerns, and a shortage of trained personnel [10,11]. Nigeria faces a significant shortage of healthcare resources, with a doctor-to-population ratio of 0.17 per 1,000, among the lowest in Africa [12]. Additionally, there are only about 250–300 radiologists, roughly one for every 658,000 people [12]. Furthermore, the country suffers from immigration of healthcare workers to developed countries [13]. This healthcare exodus deprives African countries of crucial expertise, particularly in specialties like radiology. The resulting disparities highlight the urgent need for innovative interventions to enhance early pneumonia diagnosis in children.

Artificial intelligence (AI) has made significant strides in clinical medicine and healthcare over the past decade. This era of digital medicine heralds technological advances, especially in diagnostic tools, enhancing care quality and response times [14]. Convolutional neural networks (CNNs) are a widely used AI tool in image processing and classification; renowned for their pattern detection capabilities [15]. First implemented by Fukushima in 1980 and formalized by LeCun in 1998, CNNs use convolutions, mathematical operations that allow the network to identify patterns by integrating small image regions. CNNs have shown promise in diagnosing diseases such as tuberculosis, pneumoconiosis, pneumonia, and COVID-19 [16]. Despite the emergence of newer visual transformer models [17], CNNs remain valuable due to their extensive use, simplicity, and strong research foundation.

Ayan, Karabulut [18] developed an AI paediatric pneumonia detection system using an ensemble of CNN models (VGG-16, VGG-19, ResNet-50, Inception-V3, MobileNet, SqueezeNet, Xception). Utilizing the ensemble method outperformed individual models, achieving an AUC of 95.21% and sensitivity of 97.76% on test data. Similar efforts by Labhane, Pansare [19] developed four models (basic CNN, VGG16, VGG19, InceptionV3), and used transfer learning techniques for paediatric pneumonia detection.

Furthermore, a recent systematic review by Field, Tam [20] evaluated the efficacy of AI in classifying paediatric pneumonia on chest radiographs. Their results revealed an ensemble AI achieved 96.3% sensitivity, DenseNet201 showed 94% specificity and 95% accuracy, and VGG16 had a 96.2% AUC. Some AI models nearly reached 100% diagnostic accuracy. The review emphasized AI's potential to enhance diagnosis but called for comparisons with radiologists and further clinical validation.

However, CNNs face several limitations. They require large annotated datasets, which are often unavailable in low-resource settings [21]. Their performance can drop when applied to images from different populations or protocols, raising concerns about generalizability [21]. CNNs also lack interpretability, functioning as "black boxes"; a drawback in clinical contexts that demand transparency [22].

Moreover, many of these models have scarcely been tested in clinical settings, and similar models have not been developed in Africa for childhood pneumonia detection. Therefore, this study aims to develop an AI model, using transfer learning with CNNs, to detect pneumonia in under-5 children, utilizing chest radiographs from patients in Ibadan, Nigeria.

The remainder of this paper is structured as follows: first, we present the results of the AI model; next, we discuss the findings and their implications; finally, we detail the materials and methods employed in the study.

## Results

### Data

The training set and internal test (validation) set by Kermany, Zhang [23], include 5,232 chest radiographs and 624 chest radiographs, respectively. The training set comprises 1,349 normal chest radiographs and 3,883 pneumonia chest

radiographs, while the internal test set contains 234 normal chest radiographs and 390 pneumonia chest radiographs. These details are depicted in Fig 1.

The external test set consists of 190 chest radiographs, selected according to the inclusion and exclusion criteria. Of these, 93 were normal, and 97 showed pneumonia. This distribution is illustrated in Fig 2.

## Model performance

**Training and Validation (Internal Test).** The base model and the fine-tuned model are evaluated across accuracy, precision, recall, F1 score, and area under the curve (AUC) [24]. The initial hyperparameter optimization using Optuna [25] for the base model identified the optimal settings as a maximum pooling strategy, a dropout rate of 0.25, and an initial learning rate of $4.23 \times 10^{-3}$. Subsequently, the learning rate for the fine-tuning process was found to be optimal at $2.12 \times 10^{-4}$, lower than the initial value. The model was initially trained for 6 epochs and then fine-tuned for another 8 epochs. The results from the training and internal testing (validation) of the model are presented in Fig 3.

The training loss drops sharply after the first epoch and stabilizes at a low value of 0.07. The validation loss also begins low but fluctuates, increasing slightly after the fourth epoch before returning to 0.57. Following fine-tuning, the training loss remains consistently low at 0.013. However, the validation loss rises briefly between the tenth and twelfth epochs, before a stable decline to 1.22—still, a higher value than before fine-tuning.

The accuracy plot reveals a steady improvement in training accuracy across epochs, eventually reaching close to 0.95 pre-fine tuning. Validation accuracy shows an overall upward trend, though it fluctuates significantly—starting at 0.81, dropping sharply in the fourth epoch, and recovering to 0.86 by the sixth epoch. Following fine-tuning, the training

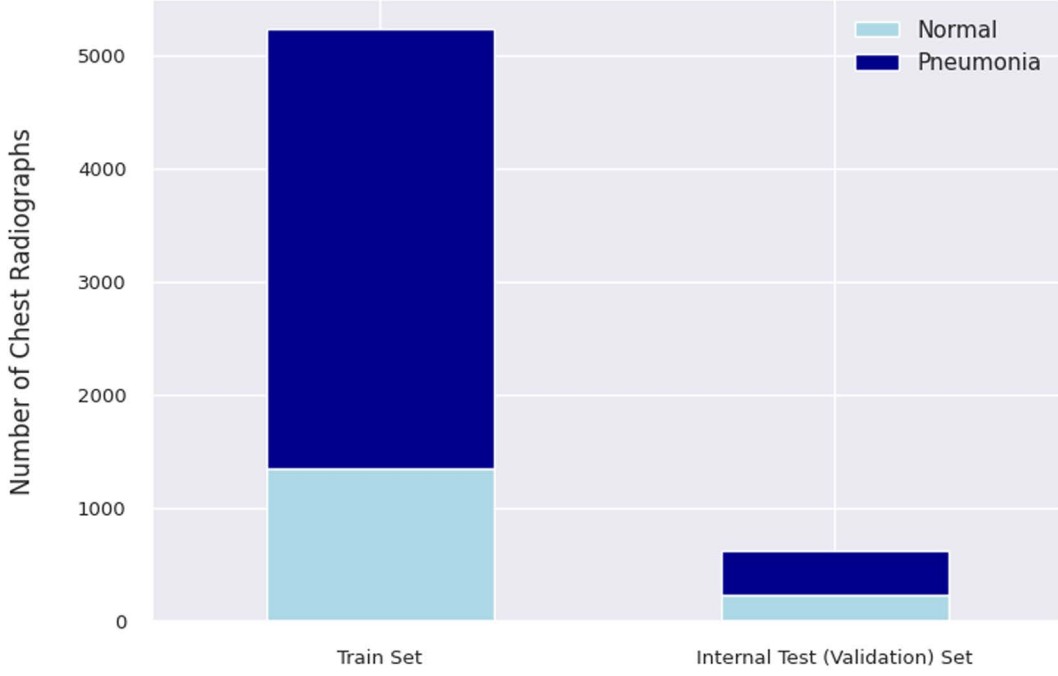

**Fig 1. Distribution of radiographs in the training and internal test sets.** *The training set includes 1,349 normal and 3,883 pneumonia chest radiographs, while the internal test set consists of 234 normal and 390 pneumonia chest radiographs.*

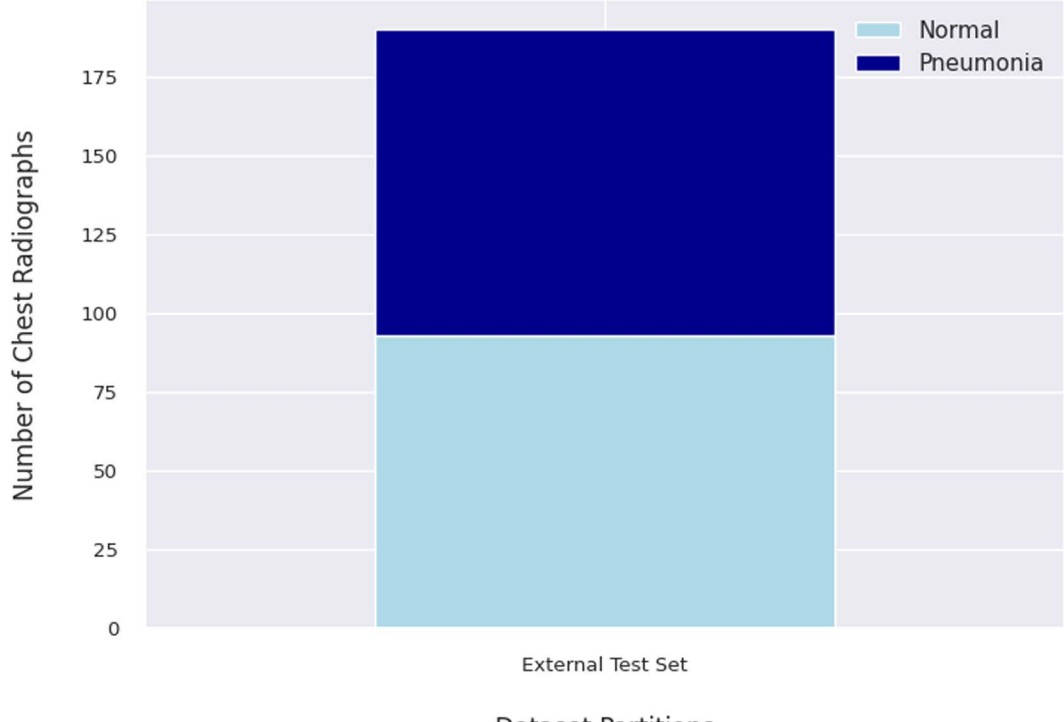

**Fig 2. Distribution of the External Test Set.** *The test set includes 93 normal chest radiographs and 97 pneumonia chest radiographs.*

accuracy remains near-perfect at 1.00, confirming the model's strong performance on the training set. However, validation accuracy remains volatile but stabilizes at 0.86 by the fourteenth epoch.

The model demonstrates strong precision during training and validation, with pre-fine-tuning scores of 0.98 and 0.83, respectively. After fine-tuning, the precision remains consistent, reaching 1.00 for training and maintaining 0.83 for validation. The lower validation precision suggests a tendency toward false positives. Recall values are stable, with pre-fine-tuning scores of 0.98 for training and 0.97 for validation, improving to 1.00 and 0.98 after fine-tuning, indicating the model effectively captures positive cases.

The F1 score, which balances precision and recall, was initially 0.85 for the training set and 0.77 for the validation set. After fine-tuning, these values improved slightly to 0.89 and 0.79, respectively. The AUC plot reflects a similar trend, with a training AUC of 0.99 and a validation AUC of 0.93 before fine-tuning, and 0.99 and 0.90 after.

**External Test.** The base model and the fine-tuned model are evaluated on the external test set. The results are presented in Table 1, comparing both performances across various metrics with 95% confidence intervals.

The base model demonstrates a lower loss (1.10) and higher accuracy (0.58) compared to the fine-tuned model, which has a loss of 3.40 and accuracy of 0.53. The F1 score is similar for both models, with the base at 0.68 and the fine-tuned at 0.67, indicating comparable balance between precision and recall. However, precision is slightly better in the base model (0.62) versus the fine-tuned model (0.59), while recall significantly drops in the fine-tuned model (0.24) compared to the base model (0.48). The area under the curve (AUC) also shows a decline from 0.65 in the base model to 0.57 in the fine-tuned model.

Further error analysis is presented through the comparison of the confusion matrices of the base and fine-tuned models (Fig 4) to assess how each model balances sensitivity and specificity, particularly in distinguishing pneumonia cases.

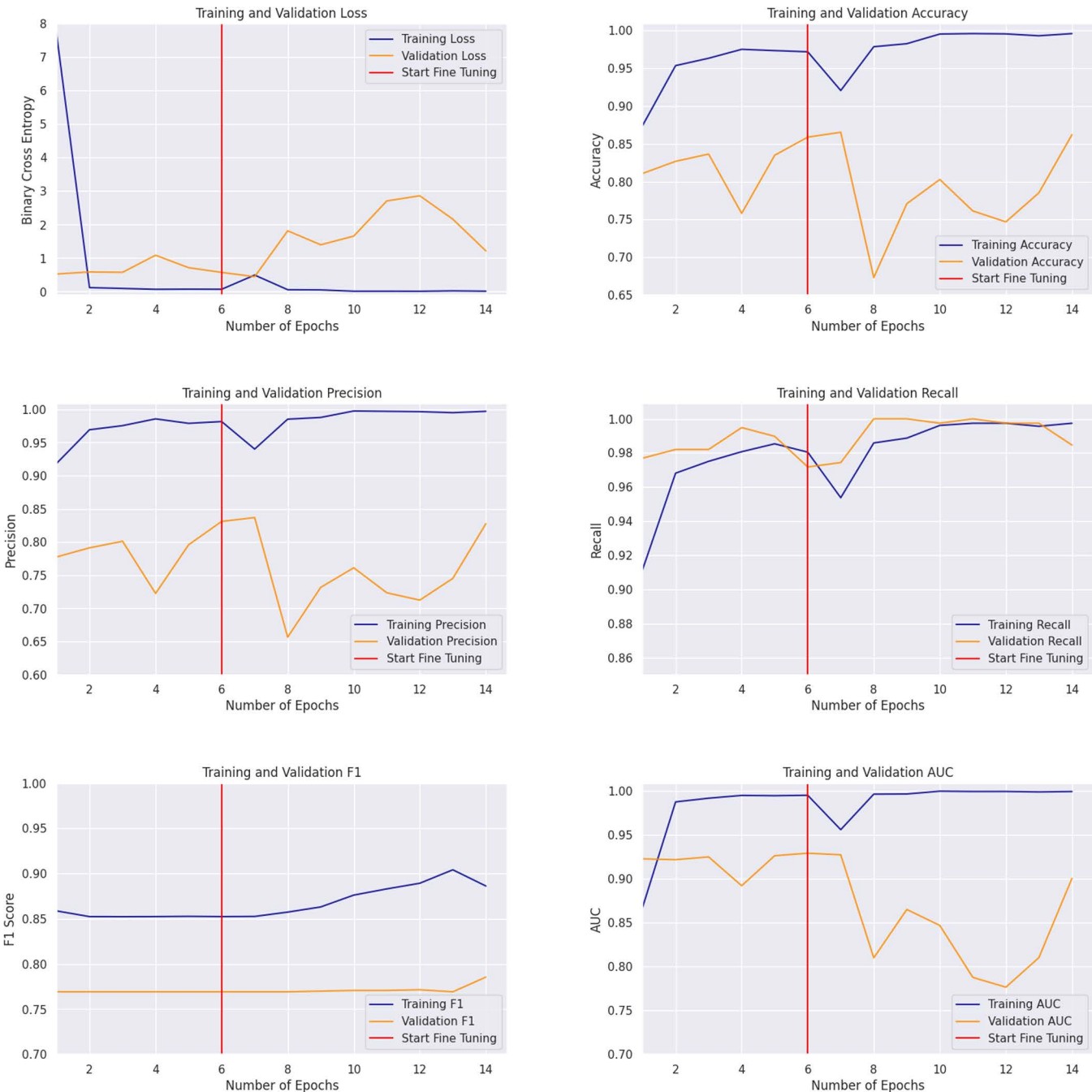

**Fig 3. Training and validation metrics over epochs: Loss, accuracy, precision, recall, F1, and AUC.** *Metrics show the model's performance trends before and after fine-tuning, across training and validation sets, providing insights into model behaviour.*

The base model correctly identified 64 out of 93 normal cases and 47 out of 97 pneumonia cases. It misclassified 29 normal radiographs as pneumonia and 50 pneumonia radiographs as normal. The overall performance of the base model demonstrated relatively balanced prediction accuracy between both categories, though it struggled more with distinguishing pneumonia cases.

**Table 1. Comparison of model performance metrics on external test set.**

|            | Loss | Accuracy | F1   | Precision | Recall | AUC  |
|------------|------|----------|------|-----------|--------|------|
| Base       | 1.10 | 0.58     | 0.68 | 0.62      | 0.48   | 0.65 |
| Fine Tuned | 3.40 | 0.53     | 0.67 | 0.59      | 0.24   | 0.57 |

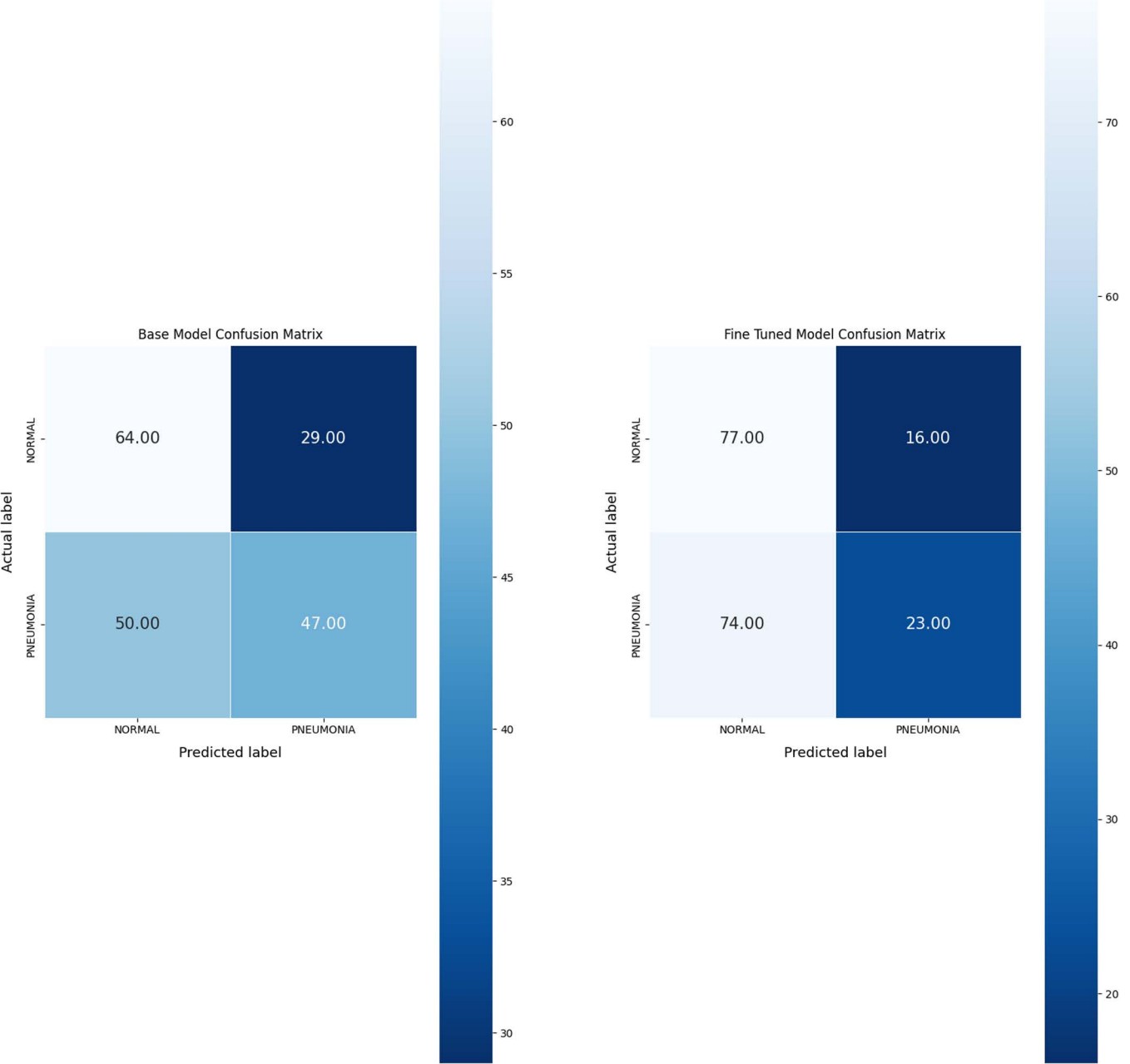

**Fig 4. Comparison of confusion matrices for base model and fine-tuned model classification performance.**

After fine-tuning, the model exhibited an improvement in identifying normal cases, correctly classifying 77 out of 93 (82.8%) normal radiographs. False positives, where normal cases were misclassified as pneumonia, decreased from 29 to 16. However, the fine-tuned model's performance in identifying pneumonia cases declined. It correctly classified only 23 out of 97 pneumonia cases (23.7%), misclassifying 74 pneumonia radiographs as normal, resulting in a notable increase in false negatives.

While the fine-tuned model improved its accuracy in identifying normal cases, it showed reduced effectiveness in detecting pneumonia, leading to an offset between sensitivity and specificity compared to the base model.

## Discussion

This study demonstrates the potential of using artificial intelligence (AI) to detect childhood pneumonia from chest radiographs in resource-limited settings like Nigeria. The model, based on the VGG-19 architecture, was able to train effectively, minimizing loss and achieving reasonable performance on internal test data. However, when tested on the external dataset from Nigeria, both the base and fine-tuned models struggled, suggesting limitations in generalizing to local image data. It is likely that differences in image acquisition processes and subtle variations, such as image resolution, between the US and Nigerian datasets may have influenced the model's performance. Image resolution is a critical factor in medical imaging, as it directly affects the visibility of fine details necessary for accurate diagnosis. Differences in pixel spacing, compression, or preprocessing between datasets can introduce domain shifts that impact model generalizability [26,27]. Studies have demonstrated that higher-resolution images improve classification performance in deep learning medical imaging tasks, while lower resolutions may lead to the loss of critical diagnostic information and reduced model accuracy [28,29]. Incorporating super-resolution techniques into the preprocessing pipeline may help compensate for resolution disparities between datasets. For example, Bastanfard, Amirkhani [30] proposed a single-image super-resolution method based on local regression and nonlocal means, which outperforms traditional techniques by enhancing fine structural details without requiring complex hardware. Applying such approaches could improve model robustness in settings where imaging quality varies significantly.

Additionally, fine-tuning did not seem to improve the model's performance, potentially due to catastrophic forgetting—a phenomenon where previously learned knowledge is overwritten when new, limited data is introduced [31]. In this case, since the VGG-19 model was pre-trained on the ImageNet dataset, which contains over 1 million non-medical images, fine-tuning on a much smaller dataset of only a few thousand images may have caused the model to 'forget' some of the general features it had learned from the larger dataset, leading to the inability to generalize well on an external dataset [32]. Importantly, the non-medical nature of ImageNet is a known drawback when applying transfer learning (TL) to medical imaging tasks, as models trained on such data may not generalize well to medical-specific image characteristics. Differences in image textures, anatomical features, and diagnostic patterns between the two domains can result in reduced accuracy when the model is tested on medical images. This critical factor contributing to the model's limited performance on external datasets is the discrepancy between the source and target domains. This may be mitigated by using a source and target domain with more similar characteristics; such as fine-tuning models on a medical-specific dataset, or using domain-adaptive training techniques, could reduce the performance gap [33,34].

One of the main strengths of this study is its pioneering approach, as it is the first to model an AI architecture for pneumonia detection in Nigeria using a large open-access dataset and testing on local imaging data. The use of transfer learning also aligns with the realities of low-resource settings, where computational and data resources are often constrained [35]. While more recent models such as DenseNet201 or vision transformers may outperform VGG-19 in accuracy, they typically require significantly more parameters, memory, and training time. In contrast, VGG-19 (with approximately 144 million parameters) offers a more computationally feasible solution for environments where advanced hardware is unavailable, making it a practical choice for pilot studies in low-resource settings. Another strength lies in the rigorous comparison between base and fine-tuned models, offering insights into the limitations of model adaptation in different environments.

However, the study has some weaknesses. The reliance on a single CNN architecture (VGG-19) may have limited the model's ability to generalize, particularly when faced with local variations in image quality and acquisition protocols. Additionally, the relatively small external test dataset may have constrained the evaluation of the model's robustness. Furthermore, statistical comparisons, such as confidence intervals and hypothesis tests, were not included in this study. This was due to the pilot nature of the study and the limited sample size of the external test dataset, which may not provide sufficient power for robust statistical analysis. Lastly, the study did not explore alternative ensemble models or newer architectures like vision transformers (VT), which could potentially perform better in heterogeneous datasets [36,37].

Compared to studies like Mabrouk, Díaz Redondo [38], which used an ensemble approach to achieve an F1-score of over 0.9, this study's use of a single model may have contributed to the lower overall performance. Gummadi, Vootla [39], who similarly employed three different architectures (VGG-16, VGG-19, and Inception ResNet V2), reported significantly higher accuracy (95.82%) and other metrics. However, these studies did not assess how well the models generalize on out-of-domain or racially different data, an aspect this study has highlighted.

Nonetheless, recent authors have continued to emphasize the problem of models performing well on internal data but failing on external dataset [40]. Glocker, Jones [41] highlighted that models built on top of the chest radiography foundation model consistently underperformed compared to the CheXpert model, resulting in differences in relative performance and raising concerns about subgroup disparities. Our study reinforces the idea that while a model may perform well during internal validation, differences in regional data may lead to significant performance drops when the model is applied to new populations.

Particularly, AI models trained on datasets from high-resource settings may not generalize well to local contexts in low-resource settings, such as Nigeria. Differences in radiomics between regions, as well as variances in imaging techniques and quality, likely contribute to this limitation. For clinicians and policymakers, this highlights the importance of localized model development and validation. Clinicians in resource-limited settings might need AI tools specifically adapted to local data to achieve reliable diagnostic support. Policymakers should prioritize the development of localized, robust, and high-quality imaging databases in Nigeria. These databases would support the development of AI models tailored to the specific needs of the population and improve diagnostic accuracy in these settings.

Several questions remain unanswered. First, the precise nature of the radiomic differences between Nigerian and US datasets requires further investigation. This would help in understanding how to better tailor AI models to local data. Additionally, the question of whether ensemble models or newer architectures, such as vision transformers (VT) [37], could better generalize to diverse datasets needs exploration. Future research should consider using ensemble models combining multiple CNN architectures or a mix of CNNs and VTs, or even exclusive ensembles of VTs, to better capture global and local image features. This should focus on evaluating the interoperability and generalization performance of newer AI architectures on diverse datasets, including those from data-sparse, low-resource regions, to draw more robust conclusions.

In addition, future work could incorporate explainable artificial intelligence (XAI) techniques to improve model transparency and clinical trust. Methods such as class activation maps (CAM), Grad-CAM, or other saliency-based visualizations can highlight the specific regions of the chest radiograph that influenced the model's decision [42,43]. This can help identify systematic sources of error by showing whether the model focused on clinically relevant lung regions or was misled by artifacts and irrelevant structures. Prior studies have demonstrated that XAI visualizations in medical imaging not only facilitate error analysis but also enable clinicians to validate AI outputs and detect biases [44]. Applying such approaches in future iterations of our model could provide valuable insight into why misclassifications occur and guide targeted improvements in model design.

Another often overlooked avenue for future work is the use of generative models, such as generative adversarial networks (GANs), to artificially augment training datasets in data-scarce settings. These models can synthesize realistic chest radiographs, potentially improving model generalization across heterogeneous datasets. Previous work

has demonstrated that GAN-generated medical images can expand dataset diversity and help mitigate overfitting in small-sample scenarios [45]. Incorporating such techniques may be particularly valuable in low-resource settings like Nigeria, where collecting large, high-quality annotated datasets is challenging.

## Conclusions

This study aimed to develop an AI model using transfer learning with CNNs to detect pneumonia in under-5 children, utilizing chest radiographs from patients in Ibadan, Nigeria. The results reveal a significant disparity in the model's performance when applied to internal versus external testing datasets, highlighting the challenges of generalizing AI models across diverse populations. This emphasizes the importance of tailoring AI solutions to specific local contexts and the critical need for high-quality, representative imaging databases in Nigeria to support independent AI development in medical imaging. Future research should also focus on building more robust and generalizable AI models to enhance their reliability and applicability across diverse populations. Addressing these challenges is essential for the successful adoption of AI technologies in clinical practice, ultimately improving diagnostic capabilities and patient care in the region. This study is limited by the use of a single CNN architecture and a relatively small external test set, which may affect the generalizability of the findings

## Methodology

This study was conducted in accordance with the recommendations outlined in the 2024 Update of the Checklist for Artificial Intelligence in Medical Imaging (CLAIM) [46].

### Study design

This is a multicenter cross-sectional study that evaluated the performance of an AI model compared to a human radiologist in differentiating radiographs of under-5 children with pneumonia (variable 1) from those without pneumonia (variable 2). The study includes both retrospective and prospective elements: the AI model is trained and internally tested on a publicly available dataset and externally tested on data collected prospectively from the study centers.

This study was designed to develop and test an AI model for diagnosing pneumonia in under-5 children, using transfer learning with pre-trained convolutional neural networks (CNNs). The goals are to explore the model's potential effectiveness in a real-world clinical setting, assessing how well it can classify radiographs of children with pneumonia and those without, addressing a significant clinical need in resource-limited environments. This exploratory aspect is crucial given the limited prior clinical validation of similar models, especially in low-resource environments like Nigeria. This allows for an investigation into the model's feasibility to provide support for radiologists in diagnosing pneumonia.

### Study setting

The study was carried out at the University College Hospital (UCH), Ibadan and Rainbow Scans Diagnostic Center, Ibadan, from November 2023 to August 2024. UCH is the oldest and largest tertiary hospital in Nigeria where several cases of childhood pneumonia are managed. UCH has board certified radiologists who possess decades of experience in paediatric imaging. Additionally, Rainbow Scans is a private medical diagnostic center also located in Ibadan; offers a range of radiological services and expertise in paediatric conditions

### Data sources

The training data consisted of a paediatric pneumonia chest radiograph dataset, which was sourced from the publicly available repository created by Kermany, Zhang [23] on the Kaggle platform. The dataset comprised chest radiographs of paediatric patients from University of California San Diego, USA, that were categorized as either normal or having pneumonia.

PLOS Digital Health

The test data comprised chest radiographs of paediatric patients aged 3 months to 5 years from the Radiology Department at the University College Hospital, Ibadan, Nigeria. This is as radiological features of pneumonia in patients below 3 months often overlap with congenital or neonatal conditions, while those older than 5 years may present with patterns resembling adult pneumonia, complicating the analysis. Additional radiographs were collected from the Rainbow Scans Diagnostic Center, Ibadan, Nigeria.

**Inclusion Criteria.** The chest radiographs with pneumonia must (1) be a confirmed radiological diagnosis of pneumonia by the consultant radiologist (2) have no complication on radiograph (such as pleural effusion, pneumatocoele etc). The variable group without pneumonia will include chest radiograph of under-5 children with normal findings.

**Exclusion Criteria.** The exclusion criteria include: (1) the chest radiographs of children older than 5 years or younger than 3 months, (2) the chest radiographs with complications of pneumonia, and (3) chest radiographs diagnosed with respiratory conditions other than pneumonia.

**Data De-identification and Preprocessing.** The chest radiographs were accessed using the RadiAnt DICOM Viewer [47] (version 2023.1, 64-bit). During this process, personal information was removed to protect patient privacy. Specifically, all identifiers, including patient names, ages, hospital numbers, and gender, were eliminated from the images. Further preprocessing was performed using the default preprocessing function specific to the model. Details of this process will be discussed in the model section.

## Image acquisition protocol

The anteroposterior projection is the alternative when posteroanterior projection cannot be done in under-5 children. The child is made to stand or sit upright or held upright depending on the age and level of cooperation. Hands are held above the head to remove the scapula from the lung fields. Immobilization devices such as the Pigg-o-stat are not available, and the accompanying adult puts on a lead jacket to protect them from scatter radiation. Images are taken when the child is calm (not crying or restless). Based on the child's age and size, the kilovolt peak (kVp) of 60–70 is used as the least setting and higher values for larger children, up to 90–125 kVp. The milliampere-seconds (mAs) is adjusted based on the child's size and body habitus. The ALARA principle (As Low As Reasonably Achievable) is adopted. Shielding, especially of the gonads, is done when necessary.

## Reference standard

The reference standard for this study was established through radiographic evaluation, determined by consensus between experienced clinicians, specifically the Chief Resident of Radiology and a Consultant Radiologist at the University College Hospital, Ibadan. Each chest radiograph is examined for typical features consistent with pneumonia, such as focal or diffuse alveolar opacities, consolidation, or interstitial infiltrates [1]. Only radiographs with clearly confirmed pneumonia and no radiological complications (such as pleural effusion or pneumatocele) are included in the pneumonia group. Normal chest radiographs are classified as those without signs of pneumonia or any other pathology.

The rationale for selecting this reference standard lies in the clinical expertise of the radiologists involved. Using a dual assessment from both a Consultant Radiologist and the Chief Resident enhances the accuracy and reliability of the diagnoses, reducing the likelihood of misclassification. While this expert-driven standard ensures high accuracy, there is a potential limitation regarding variability in interpretation. In these cases of initial disagreement, an in-house consensus review meeting is conducted to resolve discrepancies.

## Data partitions

The dataset was partitioned into training, internal test (validation), and external test sets. The training set comprised 5,232 chest radiographs, accounting for 86.54% of the total dataset. The internal validation set contained 624 chest radiographs, representing 10.32% of the dataset, and was used for model optimization. The external test set, prospectively collected,

included 190 chest radiographs, making up 3.14% of the dataset. To prevent overlap between partitions, radiographs in the external test set were disjointed at the patient level, ensuring that no patient appeared in more than one partition. This approach mitigates the risk of data leakage and maintains the independence of the external test set for unbiased performance evaluation.

### Testing data

**Sample Size Calculation and Sampling Technique** The sample size was calculated with the Leslie Fischer's formula;

$$\frac{Z^2 pq}{d^2}$$

z = appropriate value from the normal distribution for 95% confidence, 1.96
p = prevalence of pneumonia in under-5 children, 12.7% in southern Nigeria [48].
d = desired precision, 0.05.
q = 1 − p

$$\text{Sample size } (N) = \frac{1.96^2 \times 0.127 \times (1 - 0.127)}{0.05^2}$$

Hence, the estimated test sample size was 170. Using a 1:1 ratio for two variables considered, 85 chest radiographs each for subjects with pneumonia and without pneumonia (normal) are required. A cluster sampling was used to select chest radiographs of subjects with pneumonia and without pneumonia (normal) using the monthly register of chest radiographs of under-5 patients done by the Radiology Department, University College Hospital, and Rainbow Scans Diagnostics Centre. The eligible chest radiographs were selected from the dataset into clusters of pneumonia and (normal) no pneumonia. The chest radiographs were collected from November 2023 to August 2024, when the sample size was attained.

### Model development

The model in this study is based on the VGG19 architecture [49] (Fig 5) and employs transfer learning, where all layers are frozen, and then the architecture is adapted for binary classification. The inputs to the model are RGB images of size 224x224x3 with a batch size of 64, preprocessed using the "preprocess_input" function from the TensorFlow implementation of VGG19. The base VGG19 model is loaded with pre-trained ImageNet weights and includes all layers up to the final convolutional block, with no additional fully connected layers, i.e., the default classification head was not included. The model is then set up with an additional fully connected layer of 512 units with ReLU activation, and then a drop out layer before a custom binary classification head.

### Model training

The fully connected layers on top of the base model are initialized with Glorot uniform initialization [50] (the default in TensorFlow). The model is then trained with the Adam optimizer [51], with the learning rate tuned between $1e-5$ and $1e-2$. Early stopping is employed with a patience of 5 epochs, and training is limited to 20 epochs. Hyperparameter optimization is automated through Optuna [25], and spanned 25 trials. Dropout rates, pooling strategies, and learning rates are adjusted during this process to identify the best configuration. The model with the highest validation accuracy is selected. Validation accuracy serves as a benchmark for the model's expected performance on real-world, unseen data.

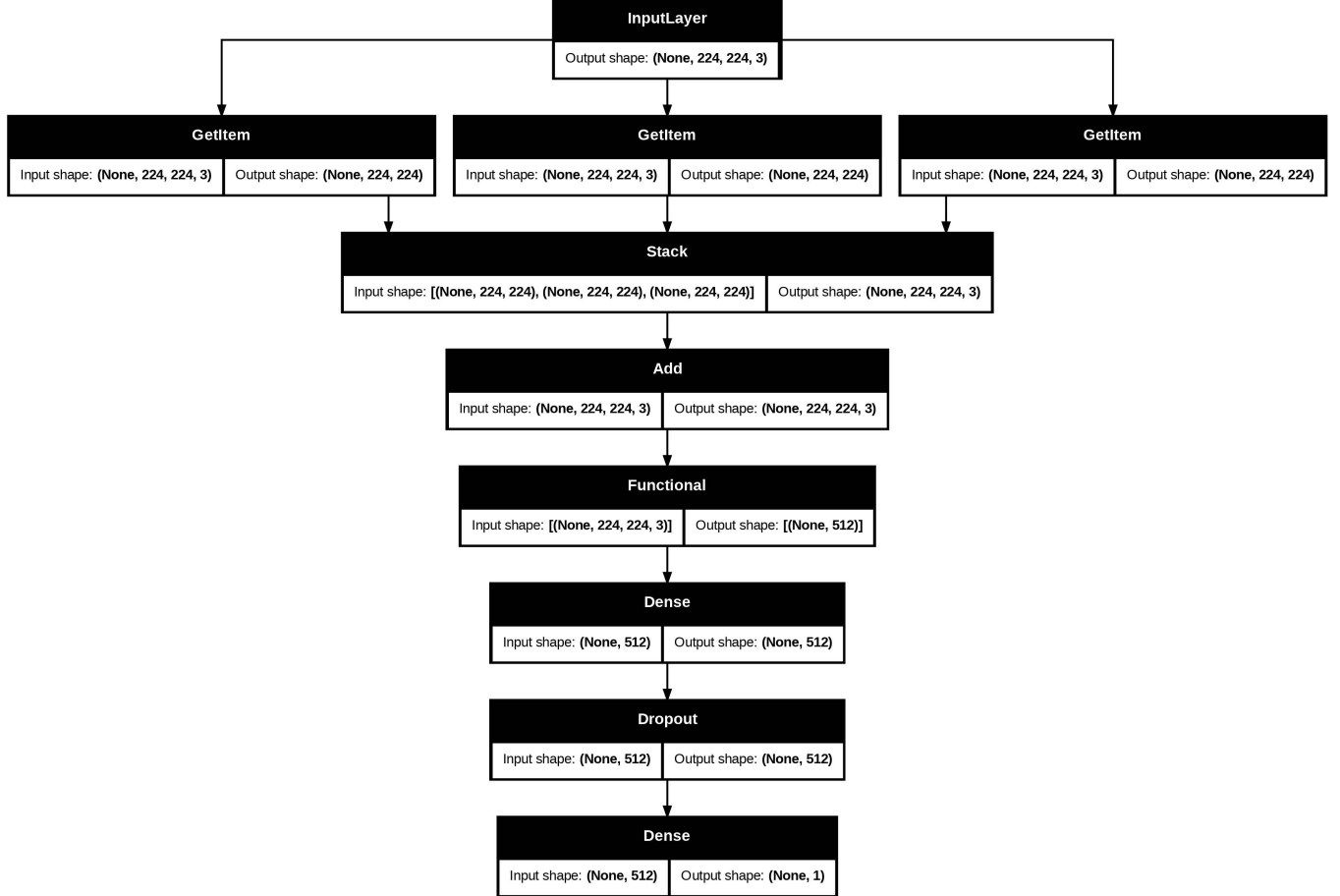

**Fig 5. Detailed Architecture of the Model.** *The neural network architecture includes an input layer for 224x224x3 preprocessed images. It utilizes a VGG19 base model with frozen weights up to the final convolutional block. Features then pass through a dense layer with ReLU units, followed by a dropout layer. The output layer, a single neuron with sigmoid activation, provides binary classification probabilities.*

## Model fine-tuning

For fine-tuning, the last 7 layers of the VGG19 model are set to be trainable, while the remaining layers remain frozen. The model is similarly configured for binary classification, using a sigmoid output to predict the likelihood of the target class [52]. Training is carried out with the Adam optimizer, with the learning rate fine-tuned between the initial best rate and one-hundredth of that value. Early stopping is applied with a patience of 7 epochs, and training was limited to a maximum of 50 epochs. Binary cross-entropy served as the loss function. Hyperparameter optimization is automated using Optuna, which conducted 50 trials to find the best parameter combinations. The learning rate is fine-tuned during this process, and similarly, the model with the highest validation accuracy was selected as the optimal version.

## Evaluation

The evaluation metrics include accuracy, precision, recall, F1 score, and area under the curve (AUC) [24]. Two evaluation rounds are conducted. First, the initial model using only transfer learning is assessed across the train, internal test, and external test sets. Next, the fine-tuned model is evaluated using the same metrics and datasets. A comparison of the results from both rounds follows.

## Supporting information

**S1 Text.** **This file provides software libraries, code repository, and links to the de-identified imaging dataset used in this study.**
(DOCX)

## Acknowledgments

We sincerely appreciate the support provided by the faculty at the Duke Center for Policy Impact in Global Health, Duke University, for their funding and mentorship. We also extend our gratitude to the leadership of the University of Ibadan Medical Students Association (UIMSA) and the College of Medicine, University of Ibadan, for their invaluable guidance and collaboration. We also acknowledge the support of the Radiology Department, University College Hospital and Rainbow Diagnostics (worthy of mention, Mrs Maryam) for granting access to the radiographs database.

## Author contributions

**Conceptualization:** Taofeeq Oluwatosin Togunwa, Abdulhammed Opeyemi Babatunde, Richard Olatunji, Godwin Ogbole, Adegoke Falade.

**Data curation:** Taofeeq Oluwatosin Togunwa, Abdulhammed Opeyemi Babatunde.

**Funding acquisition:** Taofeeq Oluwatosin Togunwa, Abdulhammed Opeyemi Babatunde.

**Investigation:** Taofeeq Oluwatosin Togunwa.

**Methodology:** Taofeeq Oluwatosin Togunwa, Abdulhammed Opeyemi Babatunde, Richard Olatunji.

**Project administration:** Oluwatosin Ebunoluwa Fatade.

**Resources:** Abdulhammed Opeyemi Babatunde, Oluwatosin Ebunoluwa Fatade, Richard Olatunji.

**Software:** Taofeeq Oluwatosin Togunwa.

**Supervision:** Oluwatosin Ebunoluwa Fatade, Richard Olatunji, Godwin Ogbole, Adegoke Falade.

**Validation:** Taofeeq Oluwatosin Togunwa, Oluwatosin Ebunoluwa Fatade, Richard Olatunji, Godwin Ogbole, Adegoke Falade.

**Visualization:** Taofeeq Oluwatosin Togunwa.

**Writing – original draft:** Taofeeq Oluwatosin Togunwa, Abdulhammed Opeyemi Babatunde.

**Writing – review & editing:** Taofeeq Oluwatosin Togunwa, Abdulhammed Opeyemi Babatunde, Oluwatosin Ebunoluwa Fatade, Richard Olatunji, Godwin Ogbole, Adegoke Falade.

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
