## [Decision Letter · Decision Letter 0]

19 Mar 2025

PDIG-D-24-00541Detection of Pneumonia in Children through Chest Radiographs using Artificial Intelligence in a Low-Resource Setting: A Pilot StudyPLOS Digital Health Dear Dr. Taofeeq Oluwatosin Togunwa, Thank you for submitting your manuscript to PLOS Digital Health. After careful consideration, we feel that it has merit but does not fully meet PLOS Digital Health's publication criteria as it currently stands. Therefore, we invite you to submit a revised version of the manuscript that addresses the points raised during the review process. Please submit your revised manuscript within 60 days May 18 2025 11:59PM. If you will need more time than this to complete your revisions, please reply to this message or contact the journal office at digitalhealth@plos.org. Please include the following items when submitting your revised manuscript:* A rebuttal letter that responds to each point raised by the editor and reviewer(s). You should upload this letter as a separate file labeled 'Response to Reviewers '. This file does not need to include responses to any formatting updates and technical items listed in the 'Journal Requirements' section below.* A marked-up copy of your manuscript that highlights changes made to the original version. You should upload this as a separate file labeled 'Revised Manuscript with Track Changes '.* An unmarked version of your revised paper without tracked changes. You should upload this as a separate file labeled 'Manuscript '. If you would like to make changes to your financial disclosure, competing interests statement, or data availability statement, please make these updates within the submission form at the time of resubmission. Guidelines for resubmitting your figure files are available below the reviewer comments at the end of this letter. We look forward to receiving your revised manuscript. Kind regards, Hadi GhasemiAcademic EditorPLOS Digital Health Hadi GhasemiAcademic EditorPLOS Digital Health Leo Anthony CeliEditor-in-ChiefPLOS Digital Healthorcid.org/0000-0001-6712-6626 **Journal Requirements:**

 **Additional Editor Comments (if provided):****Reviewers' Comments:** Reviewer's Responses to Questions

**Comments to the Author**

1. Does this manuscript meet PLOS Digital Health’s publication criteria ? Is the manuscript technically sound, and do the data support the conclusions? The manuscript must describe methodologically and ethically rigorous research with conclusions that are appropriately drawn based on the data presented.

Reviewer #1: Yes

Reviewer #2: Yes

Reviewer #3: Yes

2. Has the statistical analysis been performed appropriately and rigorously?

Reviewer #1: Yes

Reviewer #2: Yes

Reviewer #3: No

3. Have the authors made all data underlying the findings in their manuscript fully available (please refer to the Data Availability Statement at the start of the manuscript PDF file)?

Reviewer #1: Yes

Reviewer #2: No

Reviewer #3: No

4. Is the manuscript presented in an intelligible fashion and written in standard English?

Reviewer #1: Yes

Reviewer #2: No

Reviewer #3: No

5. Review Comments to the Author

Reviewer #1: The paper presents a study aimed at developing and evaluating an AI-based model for detecting pneumonia in children using chest radiographs, focusing on low-resource settings like Nigeria. Using the VGG-19 CNN architecture and transfer learning, the model was trained on a U.S. dataset and evaluated on a locally collected Nigerian dataset. Results revealed a significant performance gap between internal and external test sets, highlighting the need for AI models tailored to specific local contexts.

The study addresses an urgent health issue, pediatric pneumonia, in a resource-limited country (Nigeria), contributing to global health equity. It has a clear methodology, including a well-defined external test set and the inclusion of expert radiologists for reference standards. Although the application of computer vision models to the pneumonia diagnosis is not new, this study focuses on applying this technology to an underexplored region (Nigeria). This study acknowledges the performance gap between datasets, underscoring challenges in generalizability of AI models, contributing to broader discourse on domain adaptation. The authors also followed proper data de-identification protocols and adhered to best practices for data privacy.

Limitations of this study include a single CNN architecture (VGG-19), relatively small dataset for testing (190 images) and relatively low performance metrics (external test accuracy is 58%). The author also mentioned the shortcomings of finetuning: it failed to improve performance. Finally, this study does not discuss how the model could be integrated to clinical workflows in Nigeria.

Some recommendations are in order. The authors should consider expand their dataset. They can collect larger, more diverse local datasets to improve model robustness. Another benefit of having a larger dataset is that they can consider using some of the local data obtained in Nigeria for finetuning, potentially closing the gap of transfer learning. To improve the performance, the authors may also consider testing alternative architectures such as ensembles or vision transformers (VGG-19 is about 10 years old).

Reviewer #2: The manuscript reports detection of pneumonia in children through chest radiographs using Artificial intelligence in a low resources setting.

However, the subject is interesting, but it is too early to say any facts about the manuscript. There are some extra explanations in some parts and the other part is not clear.

Scientifically, there are several questions during reading the manuscript in my mind. Therefore, I have a few major concerns in the manuscript as followed:

_ The title should precisely declare which research question is solved, but the title of the manuscript is general. It is recommended to clearly categorize the studied work and present the research aim in a precise title.

_Abstract contains some sentences but some important aspects of abstract are missing. It should contain problems, challenges, method and evaluation clearly. It is recommended to rewrite the abstract.

_ The structure of the manuscript is not clear. The structure of the manuscript or remaining sections of the manuscript should be introduced at the end of the introduction. Please introduce manuscript sections as an outline at the end of the Introduction.

__ What are the advantages and disadvantages of the previous method? Actually, strong literature is a need for a journal article. There is some explanation about previous works. But I could not find previous works disadvantages.

_ It seems the proposed approach is mixing several works. The novel contributions are thus not clarified. Please explain where is the novelty of the proposed method.

_As it mentioned in manuscript, resolution is significant in the proposed method. In preprocessing phase, use local and global features in the proposed method as it Considered in recent work of” Toward image super-resolution based on local regression and nonlocal means. Multimed Tools Appl 81, 23473–23492 (2022). https://doi.org/10.1007/s11042-022-12584-x”.” Then compare the results for different image resolution.

_As it declared in the manuscript, computational complexity is significant. Compare computational complexity of proposed method with state of the art methods.

_What is the order of the proposed algorithm? Discuss on simplicity of algorithm and compare it with others.

_ The formula should be written in academic way in the text manuscript.

Reviewer #3: This paper is represented A Detection of Pneumonia in Children through Chest Radiographs using Artificial Intelligence in a Low-Resource Setting: A Pilot Study. There are many points that must be clarified as follow:

1. The quality of the all figures should be improved.

2. The results analysis section is very weak. Statistical results and various tests need to be examined.

3. The current structure of the paper lacks a logical and coherent flow between different sections. It is recommended to rewrite the introduction, methodology, results, and discussion sections separately and in a logically sequential manner.

4. The conclusion section requires a fundamental revision; in particular, it should explicitly address the limitations of the proposed method.

6. PLOS authors have the option to publish the peer review history of their article (what does this mean? ). If published, this will include your full peer review and any attached files.

**Do you want your identity to be public for this peer review?** For information about this choice, including consent withdrawal, please see our Privacy Policy .

Reviewer #1: **Yes: ** Yingnan Cui

Reviewer #2: **Yes: ** Azam Bastanfard

Reviewer #3: No

---

## [Decision Letter · Decision Letter 1]

16 Jul 2025

PDIG-D-24-00541R1Detection of Pneumonia in Children through Chest Radiographs using Artificial Intelligence in a Low-Resource Setting: A Pilot StudyPLOS Digital Health Dear Dr. Togunwa, Thank you for submitting your manuscript to PLOS Digital Health. After careful consideration, we feel that it has merit but does not fully meet PLOS Digital Health's publication criteria as it currently stands. Therefore, we invite you to submit a revised version of the manuscript that addresses the points raised during the review process. Please submit your revised manuscript within 30 days Aug 15 2025 11:59PM. If you will need more time than this to complete your revisions, please reply to this message or contact the journal office at digitalhealth@plos.org. Please include the following items when submitting your revised manuscript:* A rebuttal letter that responds to each point raised by the editor and reviewer(s). You should upload this letter as a separate file labeled 'Response to Reviewers '. This file does not need to include responses to any formatting updates and technical items listed in the 'Journal Requirements' section below.* A marked-up copy of your manuscript that highlights changes made to the original version. You should upload this as a separate file labeled 'Revised Manuscript with Track Changes '.* An unmarked version of your revised paper without tracked changes. You should upload this as a separate file labeled 'Manuscript '. If you would like to make changes to your financial disclosure, competing interests statement, or data availability statement, please make these updates within the submission form at the time of resubmission. Guidelines for resubmitting your figure files are available below the reviewer comments at the end of this letter. We look forward to receiving your revised manuscript. Kind regards, Hadi GhasemiAcademic EditorPLOS Digital Health Hadi GhasemiAcademic EditorPLOS Digital Health Leo Anthony CeliEditor-in-ChiefPLOS Digital Healthorcid.org/0000-0001-6712-6626  **Journal Requirements:** **Additional Editor Comments (if provided):****Reviewers' Comments:** Reviewer's Responses to Questions

**Comments to the Author**

1. If the authors have adequately addressed your comments raised in a previous round of review and you feel that this manuscript is now acceptable for publication, you may indicate that here to bypass the “Comments to the Author” section, enter your conflict of interest statement in the “Confidential to Editor” section, and submit your "Accept" recommendation.

Reviewer #4: All comments have been addressed

Reviewer #5: All comments have been addressed

2. Does this manuscript meet PLOS Digital Health’s publication criteria ? Is the manuscript technically sound, and do the data support the conclusions? The manuscript must describe methodologically and ethically rigorous research with conclusions that are appropriately drawn based on the data presented.

Reviewer #4: Yes

Reviewer #5: Yes

3. Has the statistical analysis been performed appropriately and rigorously?

Reviewer #4: Yes

Reviewer #5: Yes

4. Have the authors made all data underlying the findings in their manuscript fully available (please refer to the Data Availability Statement at the start of the manuscript PDF file)?

Reviewer #4: Yes

Reviewer #5: Yes

5. Is the manuscript presented in an intelligible fashion and written in standard English?

Reviewer #4: Yes

Reviewer #5: Yes

6. Review Comments to the Author

Reviewer #4: using an AI to assess its efficacy in LMIC is an excellent example of using AI to provide a better health care and work to reduce the health inequality. low number of data in this study is understandable and the authors have talked about the problem in both their title (pilot study) and their limitations. adopting the TL techniques to develop AI for a LMIC is an intelligence choice.

however the article needs to address some points to better show the challenges in trying to overcome this challenge.

here are some tips for authors:

1) one way to increase the number of data in your study is using Generative models to artificially increase the sample size, you can talk about such models in discussion or limitation section for future works.

https://www.sciencedirect.com/science/article/abs/pii/S1574013720303853

2) this study could use explainable techniques (XAI) to present an illustration of which part did the model mis-diagnosed and where did the most of errors occurred. please talk about this solution in discussion section.

https://www.sciencedirect.com/science/article/pii/S1566253523001148

https://link.springer.com/chapter/10.1007/978-3-031-04083-2_2

3) training on the ImageNet can also cause this error: since the its a dataset of non-medical images. this is a potential drawback for models using TL and using similar source and target domains may help reduce this error according to:

https://www.mdpi.com/2072-6694/16/11/2138,
https://www.sciencedirect.com/science/article/abs/pii/S1566253523002282

please talk more about the effect of discrepancy between source and target domain

Reviewer #5: I have reviewed the revised manuscript titled "Detection of Pneumonia in Children through Chest Radiographs using Artificial Intelligence in a Low-Resource Setting: A Pilot Study". The authors have addressed prior comments comprehensively, and the study presents valuable real-world insight into AI model performance across diverse clinical settings.

I recommend acceptance of the manuscript.

As a minor, non-mandatory suggestion, the authors may consider:

Clarifying in the discussion that statistical comparisons (e.g., confidence intervals, hypothesis tests) were not included and why.

Briefly stating the computational resources used (e.g., training time or hardware specs), to support claims about the model’s feasibility in low-resource environments.

These points are optional and do not impact my recommendation.

7. PLOS authors have the option to publish the peer review history of their article (what does this mean? ). If published, this will include your full peer review and any attached files.

**Do you want your identity to be public for this peer review?** For information about this choice, including consent withdrawal, please see our Privacy Policy .

Reviewer #4: No

Reviewer #5: **Yes: ** Morteza Atayi

---

## [Decision Letter · Decision Letter 2]

3 Sep 2025

Detection of Pneumonia in Children through Chest Radiographs using Artificial Intelligence in a Low-Resource Setting: A Pilot Study

PDIG-D-24-00541R2

Dear Dr. Togunwa,

We're pleased to inform you that your manuscript has been judged scientifically suitable for publication and will be formally accepted for publication once it meets all outstanding technical requirements.

Within one week, you'll receive an e-mail detailing the required amendments. When these have been addressed, you'll receive a formal acceptance letter and your manuscript will be scheduled for publication.

An invoice for payment will follow shortly after the formal acceptance. To ensure an efficient process, please log into Editorial Manager at https://www.editorialmanager.com/pdig/ click the 'Update My Information' link at the top of the page, and double check that your user information is up-to-date. For billing related questions, please contact billing support at https://plos.my.site.com/s/.

Kind regards,

Hadi Ghasemi

Academic Editor

PLOS Digital Health

Additional Editor Comments (optional):

Reviewer #4:

Reviewers' comments:

Reviewer's Responses to Questions

**Comments to the Author**

1. If the authors have adequately addressed your comments raised in a previous round of review and you feel that this manuscript is now acceptable for publication, you may indicate that here to bypass the “Comments to the Author” section, enter your conflict of interest statement in the “Confidential to Editor” section, and submit your "Accept" recommendation.

Reviewer #4: All comments have been addressed

2. Does this manuscript meet PLOS Digital Health’s publication criteria ? Is the manuscript technically sound, and do the data support the conclusions? The manuscript must describe methodologically and ethically rigorous research with conclusions that are appropriately drawn based on the data presented.

Reviewer #4: Partly

3. Has the statistical analysis been performed appropriately and rigorously?

Reviewer #4: Yes

4. Have the authors made all data underlying the findings in their manuscript fully available (please refer to the Data Availability Statement at the start of the manuscript PDF file)?

Reviewer #4: Yes

5. Is the manuscript presented in an intelligible fashion and written in standard English?

PLOS Digital Health does not copyedit accepted manuscripts, so the language in submitted articles must be clear, correct, and unambiguous. Any typographical or grammatical errors should be corrected at revision, so please note any specific errors here.

Reviewer #4: Yes

6. Review Comments to the Author

Please use the space provided to explain your answers to the questions above. You may also include additional comments for the author, including concerns about dual publication, research ethics, or publication ethics. (Please upload your review as an attachment if it exceeds 20,000 characters)

Reviewer #4: I appreciate the authors for their efforts to address the discussed comments of authors.

7. PLOS authors have the option to publish the peer review history of their article (what does this mean? ). If published, this will include your full peer review and any attached files.

**Do you want your identity to be public for this peer review?** For information about this choice, including consent withdrawal, please see our Privacy Policy . 

Reviewer #4: Yes: Hamidreza Ashayeri
